# Do black lives matter to employers? A combined field and natural experiment of racially disparate hiring practices in the wake of protests against police violence and racial oppression

**David S. Kirk** [ID]*, **Marti Rovira** [ID]

Department of Sociology & Nuffield College, University of Oxford, Oxford, United Kingdom

* david.kirk@sociology.ox.ac.uk

## Abstract

This study uses an experimental audit design, implemented both before and during the heightened unrest following the murder of George Floyd, to gauge the impact of Black Lives Matter and associated protests against police brutality and anti-Black racism on racially disparate hiring practices. We contrast treatment of fictitious Black and White job applicants in the labor market for service-related job openings, specifically applicants with prior experience as a police officer, firefighter, or code enforcement officer. Results reveal that the White advantage in employer call-backs and requests for an interview receded during the protests and unrest following the killing of George Floyd, even to the point of producing a Black advantage.

## Introduction

Recent high-profile cases of police violence, notably the murder of George Floyd, a Black man, by White Minneapolis police officer Derek Chauvin in May of 2020, prompted an intensity and frequency of protests and political activism against anti-Black racism arguably not seen since the civil rights era [1, 2]. Our study is motivated by an interest in understanding the political and social implications of this reckoning on racial injustice. Whereas several studies have examined the effects of police violence and resulting protests on public opinion and attitudinal change about both the police and racial injustice, we specifically seek to examine behavior change, even if temporary [3–6].

To gauge the impact of nationwide protests against police violence, and to understand the extent to which "black lives matter," in this case to employers, we conducted an audit experiment to determine if employers were any more or less likely to discriminate against Black job applicants. Audit studies are a type of field experiment used to test for discriminatory behavior, typically in studies of employment or housing discrimination [7]. Whereas an expansive empirical literature has developed over the past two decades using audit and related correspondence experiments to examine employment discrimination [8–10], only recently have a

**Data Availability Statement:** Our study is registered on Open Science Framework (https://osf.io/p8je9/registrations), where our data and Stata code is archived.

**Funding:** This work was supported by the British Academy (PF19\100020 to MR), the John Fell Fund of the Oxford University Press (to DK), and the Leverhulme Trust through the Leverhulme Centre for Demographic Science (to DK). The funders did not play any role in the study design, data collection and analysis, the decision to publish, or the preparation of the manuscript.

**Competing interests:** The authors have declared that no competing interests exist.

small number of studies similar to ours leveraged a natural experiment with an audit experiment to examine how rates of employment discrimination may change in response to policy or exogenous shocks [11–13]. However, none of these studies of employment discrimination have examined the context of an exogenous shock associated with police violence and resulting social protests against racial injustice.

We implemented our audit experiment in two waves, before and then during the heightened unrest associated with the murder of George Floyd. For our experiment, we submitted fictitious résumés to real service-related job openings posted on Indeed.com and Craigslist, using racially distinctive names to signal race, and then recorded whether a given job applicant received an affirmative response from the employer to interview for the position or to discuss the job (via a voicemail, text, or email). Analyses to follow focus on whether the gap in response rates between Black and White individuals varied significantly from the period prior to Mr. Floyd's murder to the period immediately following his murder.

## Theoretical background

Our study of the effects of political protest in response to lethal police violence is grounded in two literatures. First, our study is informed and motivated by work on the role of the police as an instrument of oppression against ethnic communities [14, 15]. As social scientists have importantly ascertained, in many marginalized communities, the police are the face of the government, and interactions with the police represent the primary way residents interact with the state. The protests in the wake of George Floyd's murder were a response not merely to the murder, but to the continued marginalization and subjugation of Black communities by state actors, particularly the police [15]. Our contribution to this literature is to examine whether the occurrence of George Floyd's murder, and the mobilization against state oppression and racial injustice that followed, yielded gains for marginalized communities, in this case in the realm of employment.

Second, our study is grounded in sociological theorizing on race relations and racial prejudice, particularly work by Herbert Blumer [16, 17]. In his group threat theory, Blumer (p.4) characterizes four feelings at root of prejudice by a dominant group: (1) a feeling of superiority, (2) a feeling that the subordinate race is intrinsically different and alien, (3) a feeling of proprietary claim to certain areas of privilege and advantage, and (4) a fear and suspicion that the subordinate race harbors designs on the prerogatives of the dominant race [16]. These feelings can help us disentangle whether to expect progress on race relations following large-scale social protests, or perhaps a backlash, as has been a common narrative about the fallout from the civil rights protests of the 1960s [5, 18, 19].

To Blumer, prejudice is relational, and derives from a perceived challenge to group position. Importantly, but perhaps obviously, perceived group position is not static. Because group position and race relations are dynamic, Blumer theorized a process of "collective definition" to understand the ways in which groups form views about relative group positions of dominant and subordinate groups [16, 17]. Quillian and Midtbøen characterize Blumer's theory as a situational theory of discrimination, and suggest that the extent of discrimination against an outgroup depends on contemporary social forces [7]. Pertinent to the current study, Blumer and Duster (p.131) argue that this process of assessment and reassessment of race relations is "profoundly influenced by highly-dramatic events," such as "well-publicized brutal police action against members of a racial group" [17]. Prompted by police violence and a rising consciousness of stark racial inequalities in society, superordinate groups may be confronted by an antagonistic pair of forces, one involving action to maintain social advantages through exclusion practices and the other a "gate-opening orientation" in which the barriers to an equal and equitable society are lowered, even if temporarily [17]. For instance, increasing

awareness of racial injustices may lead superordinate groups to reconsider its "feeling of proprietary claim to certain areas of privilege and advantage" (p.4), thereby opening up access to subordinate groups to some of the advantages of their position [16]. Indeed, one reason we draw upon group threat theory to situate our study over other explanations for employment discrimination (e.g., racial prejudice, implicit bias, organizational and institutional theories) is that group threat may be able to explain relatively abrupt shifts in discriminatory practices whereas other perspectives may predict slower change.

Despite the appeal of group threat theory for examining discrimination in the wake of exogenous shocks associated with ethnic relations, a recent study of the potential reduction in employment discrimination against Muslims casts doubt on whether highly-dramatic events can really lower barriers to a more equitable society. In their study of the anti-Muslim terrorist attacks by Anders Breivik in Norway in 2011, Birkelund and colleagues use two waves of an audit experiment to determine if employment discrimination against individuals of Pakistani descent declined as solidarity with Muslims purportedly grew following the attacks [12]. However, they do not find evidence that employment discrimination against Pakistanis changed from before to three to five months after the attack. The authors acknowledge, however, that they were unable to examine labor market practices immediately after the terrorist attack (i.e., less than three months), so it possible that there may have been a very short-term reduction in employment discrimination against Pakistani job applicants, which had reverted to pre-existing levels within three months.

Whereas our research and the study by Birkelund and colleagues both examine the potential "gate-opening" effect of dramatic events on ethnic discrimination in hiring practices, research reveals support for the counter situation, where a threatening event reinforces racially discriminatory hiring practices [12]. Specifically, Mobasseri has shown how exposure to violent crimes occurring proximate to employers exacerbates disparate hiring practices against Black job applicants [13]. While both the Mobasseri and Birkelund et al. studies are highly informative, it remains to be seen if recent protests in response to police violence in the United States yield the type of gate-opening effect theorized by Blumer and Duster [12, 13, 16].

The foregoing discussion leads us to the following hypothesis. We expect that in the period immediately after George Floyd's murder and the eruption of protests nationwide, employment discrimination against Black applicants declined relative to the period prior to Mr. Floyd's death. The basis of our expectation is that the murder and protests sparked a reckoning on race relations as well as rising support for Black Lives Matter and an increased perception of anti-Black discrimination among some groups [6, 20]. In turn, employers consciously or unconsciously reduced barriers to employment opportunities for Black applicants.

## Methods

In this study, we combine a randomized audit experiment and natural experiment to examine the extent to which employment discrimination at the point of hiring is malleable, in this case in response to an exogenous shock associated with widely publicized police violence. Our design is a version of an audit experiment known as an online correspondence test, in which we applied to online job advertisements.

We submitted job applications in two time periods: (1) prior to the murder of George Floyd (May 2019 to March 2020, $N$ = 1210) and (2) immediately following Mr. Floyd's murder (June 4, 2020 to July 16, 2020, $N$ = 424). The combination of the size, intensity, and frequency of protests against police brutality and racial injustice following Mr. Floyd's murder were, according to experts, "unprecedented," underscoring the potential for differential treatment of job applicants by race in the second time period relative to the first [1].

Whereas we randomly assigned the race of applicants to job openings, applications were unmatched by race, such that Black and White candidates were not applying to the same jobs. Hence, in contrast to many audit studies of racially discriminatory employment practices, in which one Black auditor and one White auditor apply for the same position, we do not employ a matched-pairs design by race (although, as described in the next subsection, we do use a matched design by prior profession). This decision to not use a paired design by race was based on resource constraints and ethical reasons, in that we sought to minimize the number of jobs to which we applied. We note that methodological work on audit designs suggests that in some cases, unmatched designs may be more statistically efficient than matched designs [21, 22].

## Prior profession

We constructed the fictitious résumés so that an applicant's most recent profession was as a police officer, firefighter, or code enforcement officer, with approximately three years of experience in these professions and departure from the job within the month preceding the application for a new job (with the reason for the departure unspecified). Our motivation for listing these professions on our résumés was to examine whether certain occupational choices may be stigmatized in the labor market, particularly when an applicant seeks to switch careers. Given the ongoing challenge of lethal police violence in the United States, our larger project is partially focused on the potential stigmatization of the police in the labor market. For instance, some employers may view experience as a police officer as a signal that the individual holds prejudicial views or may be overly aggressive and controlling. Whereas our study was motivated from the start to examine the potential stigma of policing, we certainly could not anticipate the tragic events in Minneapolis or the magnitude of the protest activity that followed.

The focus on policing as a prior profession allows us to disentangle whether any changes over time in employer responses to Black and White job applicants reflects changing levels of discrimination against Black applicants, or perhaps more reluctance and declining preferences for hiring particular White applicants (i.e., White police officers). We chose firefighting and code enforcement for comparison occupational backgrounds given their similarity in skillsets to police officers, at least with respect to the applicability of skills for our sampled jobs. That being said, in a forthcoming research note using the same data employed here, we do not find evidence that former police officers are treated any differently in the labor market following widespread protests against police violence, although that article does not focus on the question of race differences in employment [23].

When applying for each given job, we submitted job applications from two applicants, one a former police officer paired with a second application from either a former firefighter or code enforcement officer. Candidates for a particular job had the same racial background (i.e., both applicants were either White or Black) and gender, and both race and gender were randomly assigned to job openings. The number of job applications by prior profession, race, and gender can be seen in Table 1.

## Sampled metropolitan areas and block randomization

We sampled employers in two Northeastern metropolitan areas, Boston and Philadelphia, effectively implementing both waves of the experiment separately in blocks defined by location. We originally intended to sample employers from two metro areas in each region of the country in order to more fully assess whether racial gaps in employer responses varied by city and region, but resource constraints and the onset of the Covid-19 pandemic ultimately prevented us from extending the study beyond the Northeast. Using data from the Fatal

**Table 1. Number of observations by time period, applicant race, gender, and prior profession.**

|  |  | Number of Obs. | |
| --- | --- | --- | --- |
| **Profile** |  | **Pre-Floyd Murder** | **Post-Floyd Murder** |
| Black Male Police |  | 140 | 48 |
|  | Black Male Firefigher | 82 | 25 |
|  | Black Male Code Enf. | 58 | 23 |
| White Male Police |  | 141 | 65 |
|  | White Male Firefigher | 65 | 38 |
|  | White Male Code Enf. | 76 | 27 |
| Black Female Police |  | 153 | 48 |
|  | Black Female Firefigher | 77 | 18 |
|  | Black Female Code Enf. | 76 | 30 |
| White Female Police |  | 171 | 51 |
|  | White Female Firefigher | 84 | 31 |
|  | White Female Code Enf. | 87 | 20 |
| Total Job Postings |  | 605 | 212 |
| Total Applications |  | 1210 | 424 |

Encounters repository on police killings (http://www.fatalencounters.org/), we categorized primary cities in each of the metro areas in the Northeast with more than one million population as either having low or high rates of police violence. We did so by ranking the cities based on the rate of deaths by the police from 2013 to 2017 and splitting the list at the median. We then randomly selected one metro area to sample from the bottom half (Boston) and one from the top half (Philadelphia) of the distribution in terms of rates of police violence. We only sampled large metro areas in order to ensure a sufficiently large number of potential jobs for which to apply, and to avoid diluting the job market with fictitious applications.

## Selection of names

Whereas our broader project included a focus on potential stigma from a prior profession, our primary focus in the current study is disparate employer responses to applicant race. We signaled race through first and last names used on résumés, on online job application forms, and in emailed applications, as follows:

○ Black male: Jabari Washington (prior experience as a police officer), Tremayne Jefferson (firefighter), and Darnell Mosley (code enforcement officer);

○ White male: Ethan Becker (police), Ryan Walsh (fire), and Jake Meyer (code enforcement);

○ Black female: Shanice Jefferson (police), Erykah Jackson (fire), and Janae Booker (code enforcement);

○ White female: Claire McGrath (police), Emily Decker (fire), and Katelyn Hartman (code enforcement).

Our selection of putatively Black and White names was guided by prior methodological work on audit studies, particularly work by Gaddis [24]. The extent to which these names solely signal race rather than additional characteristics such as socioeconomic status is an important consideration for assessing the validity of our inferences about racial discrimination [24, 25]. Accordingly, we provide a detailed rationale for our selection of these names in our S1 File.

## Characteristics of fictitious applicants

We developed applicant profiles and corresponding résumés to ensure similarity across background characteristics, but without exactly duplicating pieces of information or résumé style. For educational background, applicants had graduated high school, but with no further education thereafter. Prior to becoming a police officer, firefighter, or code enforcement officer, our fictitious applicants worked as delivery drivers and as sales clerks/cashiers in retail businesses, for approximately 2.5 years total. Combined with 3 years of experience as police officers, firefighters, or code enforcement officers, our applicants had about 5.5 years of work experience following high school. Specific elements of the résumés (e.g., addresses, high school attended, graduation month, prior jobs, and skills) were randomized for each job opening.

## Characteristics of targeted jobs and employers

We submitted fictitious résumés to four different categories of service-related jobs posted on Indeed.com and Craigslist.com: (1) skilled trades, (2) transportation, 3) sales, and (4) office and customer support. We expected that for these four occupations, former police officers, firefighters, and code enforcement officers would be similarly qualified, as would individuals of different racial groups. Where information was available on job postings, we targeted jobs with a salary approximately 1.5 times the living wage in the metropolitan area.

 We sampled newly listed job openings occurring within the preceding week. We did not apply to jobs requiring a social security number, nor did we sample jobs requiring an immediate online qualification, psychological test, or automated video interview. We only applied to one job opening per employer. For companies with multiple branches and sites, we only applied to one position per branch. We sampled jobs on Indeed.com and Craigslist by convenience, as opposed to a random sample of all recently posted job openings.

 As noted, when applying for jobs we submitted job applications from two applicants, one a former police officer and then either a former firefighter or code enforcement officer. We randomized the order of the application—i.e., whether the first application was sent from the former police officer or the firefighter/code enforcement officer. We submitted the second application on the same day, approximately three to five hours after the first submission.

## Randomization procedure

As described in preceding paragraphs, we used randomization across several aspects of our experimental design. To summarize, we randomized: (1) the information placed on résumés and cover letters, (2) the control condition used for a given job (i.e., firefighter or code enforcement officer), (3) the order of the paired applications (i.e., whether the application from the former police officer was first or second), and (4) the demographic profile of the applicants for a given job (i.e., Black males, Black females, White males, or White females). Specifically, for each month of data collection, we used Lahey and Beasley's *Resume Randomizer* program to create 100 different batches of résumés for each demographic combination of fictitious job applicants (i.e., Black male, Black female, White male, and White female) and metropolitan area (Boston and Philadelphia) [26]. For each batch, we created two résumés, one for a police officer and another for a control condition (firefighter or code enforcement officer). Which control condition to include in the batch was assigned randomly by the *Resume Randomizer* program. We randomly assigned which batch (i.e., from 1 to 100) of the paired résumés to use when applying for a particular job using the Excel function *RANDBETWEEN*. We also randomly assigned the order of applications (i.e., whether the former police officer applied for the job first or second) using the *RANDBETWEEN* function.

## Outcome variable

Each experimental profile had an email account and a unique phone number, which could receive texts and voicemails. Our binary outcome variable measures whether the fictitious applicant received an affirmative response from the employer to set-up an interview or to discuss the job opening. We excluded immediate auto-generated responses acknowledging receipt of the job application.

## Estimation

For estimation, we use linear probability models (LPMs). For probabilities in the range of 0.20 to 0.80, as is generally the case of the callback rates in our audit study, the LPM is a very close approximation to the logistic model, with both models fitting similarly well [27]. In this case, one might favor the LPM, as we do here, because of its ease of interpretation [28, 29]. Beyond ease of interpretation, another reason to estimate LPMs is to facilitate examination of possible heterogeneous effects by time period and by prior profession [29]. In a logit or probit model, interaction effects are conditional on other independent variables, undermining clear interpretation. Nevertheless, we replicated our analyses using a logit model with highly similar results, which can be found in S1 Table in S1 File.

For the analysis, our data is structured such that there is one record per applicant. As two applicants applied for each position, a police officer and either a firefighter or code enforcement officer, we constructed a job ID indicator that is duplicated across the two applicants who applied for a given job. In our LPMs, we cluster our standard errors by these paired job IDs.

Models include the location of the job (Boston vs. Philadelphia), the online site advertising the job (Craigslist vs. Indeed.com), job type (skilled trades, transportation, sales, and office and customer support), and month and day of the week of the job application as control variables, in addition to treatment and control variables already mentioned (i.e., race, time period, prior profession, and gender).

## Methodological limitations

It is important to recognize that the time duration of our audit experiment overlaps the Covid-19 pandemic. Because of the pandemic, the labor market in the year leading up to Mr. Floyd's death (i.e., our first time period) was dissimilar to the labor market after his death (i.e., time period 2). For instance, according to the Bureau of Labor Statistics, the average unemployment rate in Boston was 2.65% in the first time period of our study (May 2019 to March 2020), and 5.85% in Philadelphia. In the second time period, June and July 2020, the average unemployment rate was 12.1% in Boston and 18.95% in Philadelphia. Hence, we suggest that comparisons related to the *level of employment* across time periods be interpreted cautiously.

However, our emphasis in the study is on the White-Black difference in employer callbacks, rather than on the level of employer responses. Research suggests that the costs to employers for discriminating will be higher in a tight labor market with few suitable job applicants [30, 31]. Conversely, there may be less cost to discriminating employers in a slack labor market. Hence, we might expect that during the early stages of the pandemic, employers would have been more likely to discriminate against Black applicants given slack in the market. Should we find that the White-Black gap in employer call-backs narrowed following the protests from Mr. Floyd's murder, the reduction may be less than in might have been in a "normal" labor market not characterized by pandemic-induced slack.

A possible limitation of our study is the convenience sampling strategy we employed. We sampled recently posted jobs on Indeed.com and Craigslist by convenience rather than

conducting a random sample of all recently posted job openings. Moreover, our design focuses on entry-level service-related jobs, particularly those with a routine demand for new employees [32]. These jobs represent just a fraction of the labor market, and the nature of recruitment for these jobs may result in typically lower levels of discrimination than in other sectors. For instance, per our discussion above about slack vs. tight labor markets, given the frequent demand for qualified workers in the skilled trades, delivery and transportation, sales, and office and customer support, discrimination rates may be lower than with other jobs because there is often a shortage of workers. The implication is that in various ways, our results may not fully generalize to the population of jobs in a labor market. Of course, this limitation is true across all audit studies of labor market discrimination.

Finally, our use of applicants with backgrounds as police officers, firefighters, and code enforcement officers may be regarded as both a strength and limitation of our design. On the one hand, the inclusion of police officers by race presents us the opportunity to determine if, for example, there are general negative reactions to police violence against all police officers, or if any reactions are limited to White officers given the context of Mr. Floyd's death by a White officer. On the other hand, we are unable to determine if the findings of our study generalize beyond the narrow professional background characteristics we include in the study. We acknowledge the trade-off and potential limitation of this design choice.

### Ethics approval

This project was approved by the University of Oxford's Central University Research Ethics Committee (R61595/RE001 & RE002). Nevertheless, additional comments on the ethics of the experiment are warranted given debates in the literature about the ethics of correspondence tests [33, 34].

By design, correspondence studies of discrimination involve deception, given that participants, employers in our case, are not made aware that they are participating in an experiment. Therefore, we did not obtain informed consent from the employers who participated in the study. There are trade-offs between the costs and benefits of audit and correspondence designs, but the use of deceptive study designs may be acceptable under the following conditions: a) other research designs (i.e., that do not use deception) cannot similarly overcome the methodological obstacles to measuring discrimination; b) the topic has social relevance; and c) there is minimal harm to participants and minimal negative externalities [34–36]. We suggest that our study meets these conditions, and in our S1 File we provide a detailed justification for our use of deception.

### Results

Our results reveal that prior to the killing of George Floyd, 24.5 percent of White job applicants in Boston and Philadelphia received a call, text, or email response from sampled employers to discuss the job opening or to interview for the position, in comparison to 17.9 percent of otherwise similar Black applicants, for a statistically significant 6.6 percentage point disparity (see Fig 1, with corresponding coefficients and standard errors in Table 2). The White-to-Black ratio in terms of the call-back rate is 1.37 (24.5 / 17.9), which is almost exactly equivalent to the disparity reported in a recent meta-analysis of audit studies by Quillian and colleagues [8]. In that study, the authors find no change in the levels of employment discrimination against Black applicants in the US between 1989 and 2015, and that White applicants received, on average, 36% more callbacks from prospective employers than otherwise similarly qualified Black applicants. Hence, whereas our focus is specifically on entry-level job applicants with a prior background in policing, firefighting, or code enforcement, we observe a pattern of

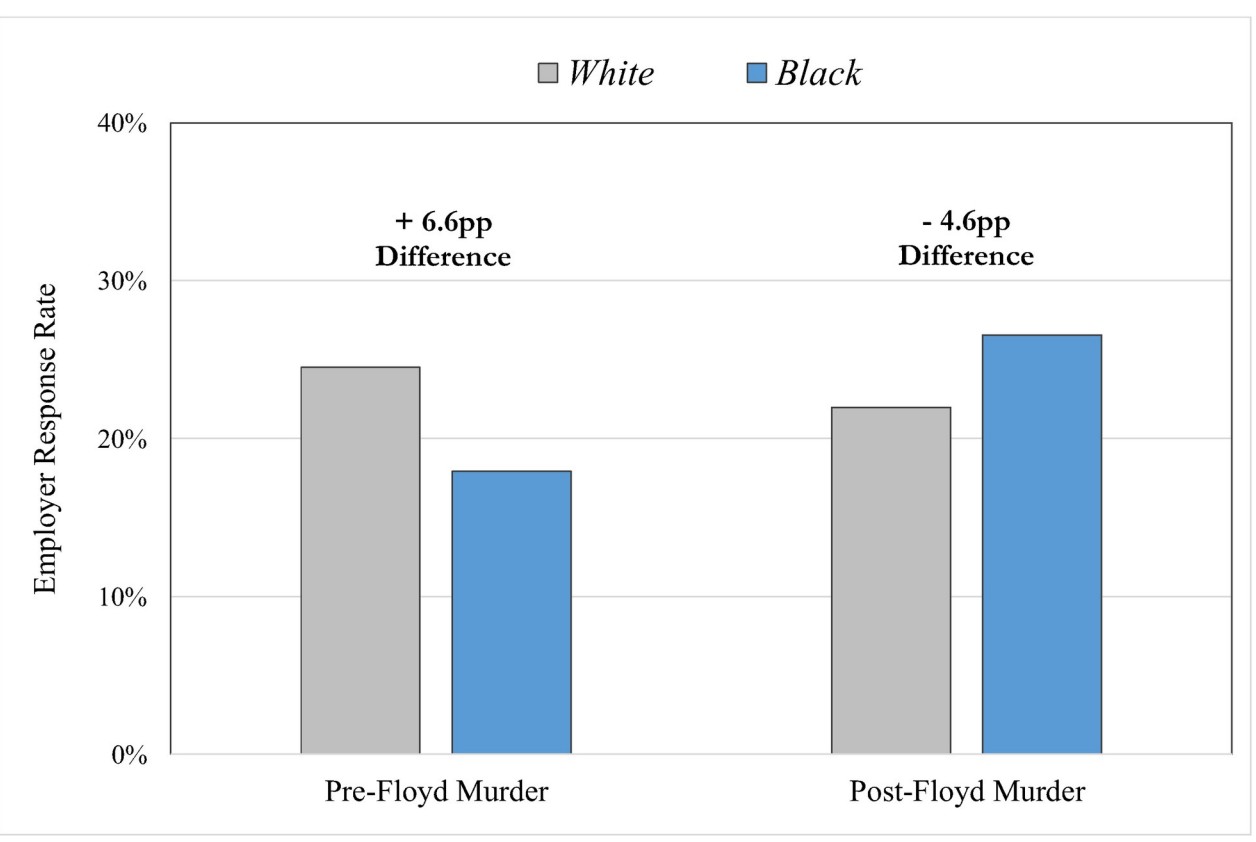

**Fig 1. Results from two time periods of an audit experiment to gauge employer responses to race among otherwise similar fictitious job applicants.** The figure displays a cross-tabulation of response rates, by race and time period. Results in the figure also correspond to the predictive margins of the interaction between race and time period from Model 1 in Table 2.

employment discrimination similar to other audit studies of employment discrimination in the United States.

Where our study moves beyond extant research is the replication of the design in the aftermath of highly publicized police violence. In the six-week period immediately following George Floyd's murder, the racial gap in employer responses reversed to a 4.6 percentage point advantage for Black applicants (22 percent vs. 26.6 percent; White-to-Black ratio of 0.83). The corresponding interaction term in Model 1 of Table 2 indicating the 11.2 percentage point shift between the White advantage in time period 1 (i.e., 6.6pp) to the Black advantage in time period 2 (4.6pp) is marginally significant (p = 0.075).

In Model 2 we add controls for gender, location of the job, online job platform, job type, prior profession, and the month and day of the week of the application, with results similar to estimates from Model 1 in terms of the race by time period interactions. We also see that employer callbacks for jobs related to office and customer support as well as sales were less likely relative to driver positions.

In Model 3 we add three-way interactions between race, time period, and prior profession, to determine if the changes in the racial gap in employer responses over time is more pronounced among one particular profession. For instance, given the context of the study—the killing of a Black man by a White police officer—it is plausible that reversal of the White advantage in employment depicted in Fig 1 reflects a declining preference for hiring White police officers rather than an increasing preference for Black job applicants irrespective of

**Table 2. Linear probability models of employer responsiveness to job applications, by time period and applicant race.**

|  | Model 1 | | | Model 2 | | | Model 3 | | |
|---|---|---|---|---|---|---|---|---|---|
|  | Coef | (SE) |  | Coef | (SE) |  | Coef | (SE) |  |
| Constant | 0.179 | (0.020) | *** | 0.296 | (0.079) | *** | 0.302 | (0.079) | *** |
| Race (White) | 0.066 | (0.030) | * | 0.060 | (0.029) | * | 0.046 | (0.033) |  |
| Time Period 2 (Post-Floyd) | 0.086 | (0.046) | + | 0.057 | (0.055) |  | 0.056 | (0.059) |  |
| Race x Time Period 2 | -0.112 | (0.063) | + | -0.111 | (0.062) | + | -0.113 | (0.068) | + |
| Gender (Male) |  |  |  | 0.022 | (0.026) |  | 0.023 | (0.026) |  |
| Boston (vs. Philadelphia) |  |  |  | 0.035 | (0.030) |  | 0.035 | (0.030) |  |
| Craigslist (vs. Indeed.com) |  |  |  | -0.022 | (0.037) |  | -0.021 | (0.037) |  |
| Office/Cust. Support Job (vs. Driver) |  |  |  | -0.126 | (0.041) | ** | -0.126 | (0.041) | ** |
| Sales Job (vs. Driver) |  |  |  | -0.037 | (0.043) |  | -0.036 | (0.043) |  |
| Skilled Trades Job (vs. Driver) |  |  |  | -0.167 | (0.042) | *** | -0.168 | (0.042) | *** |
| Firefigher (vs. Police) |  |  |  | 0.018 | (0.018) |  | 0.011 | (0.028) |  |
| Code Enf. (vs. Police) |  |  |  | -0.009 | (0.019) |  | -0.035 | (0.031) |  |
| Race x Firefighter |  |  |  |  |  |  | 0.010 | (0.043) |  |
| Race x Code Enf. |  |  |  |  |  |  | 0.047 | (0.042) |  |
| Time Period 2 x Firefighter |  |  |  |  |  |  | -0.050 | (0.066) |  |
| Time Period 2 x Code Enf. |  |  |  |  |  |  | 0.048 | (0.067) |  |
| Race x Time Period 2 x Firefighter |  |  |  |  |  |  | 0.093 | (0.082) |  |
| Race x Time Period 2 x Code Enf. |  |  |  |  |  |  | -0.090 | (0.090) |  |
|  |  |  |  |  |  |  |  |  |  |
| Month indicator included | NO | | | YES | | | YES | | |
| Day of week indicator included | NO | | | YES | | | YES | | |

'+ p<0.10

* p<0.05

** p<0.01

*** p<0.001 (two-tailed test).

N = 1,634 job applications (817 jobs). Standard errors clustered by job.

Marginal effects from Model 1 estimates are displayed in Fig 1. To condense the presentation of results, we have omitted from the table the coefficients and standard errors for the indicators of the month and day of the week when job applications were submitted.

prior profession. However, the lack of statistically significant three-way interactions in Model 3 (i.e., the Race x Time Period 2 x Firefighter coefficient or the Race x Time Period 2 x Code Enforcement coefficient) suggests that the reversal of the White advantage in employer responses from time period 1 to period 2 reflects lessening discrimination against Black applicants rather than declining employer preferences for White former police officers.

## Discussion

Whereas a select number of public opinion polls and empirical studies reveal that attitudes about police and perceptions of racial discrimination changed following George Floyd's murder and the resulting Black Lives Matter protests [6, 20], our contribution has been to focus on tangible changes in discriminatory behavior, in this case in the realm of employment.

Our results reveal that, for applicants with prior backgrounds as front-line public sector employees, the atmosphere of unrest around police brutality and anti-Black racism following the murder of George Floyd reduced levels of discrimination in the service-related labor market. This observed shift in racially disparate hiring practices after a highly-dramatic event is consistent with Blumer and Duster's theorizing about group threat [17]. The death of Mr.

Floyd and the protests that immediately ensued may have raised consciousness about racial inequalities among employers, leading some to reduce exclusionary practices against Black job applicants.

For how long employment practices remain more equitable as societal attention to racial injustice wanes is a critical question, although not one we are able to answer in the present study. Social psychological research on interventions to alter implicit racial biases suggests that while preferences for some racial groups may be malleable in the very short term, biases tend to be quite stable in the long-run [37, 38]. Consistent with these studies, Birkelund et al. did not find evidence of a sustained change in ethnic discrimination against Pakistani job applicants after an anti-Muslim terrorist attack in Norway [12]. Moreover, recent meta-analyses of audit experiments, in both the United States and Britain, reveal that levels of hiring discrimination against ethnic minorities tend to remain stubbornly persistent over many decades [8, 9].

Hence, it may be the case that reduced discrimination toward Black applicants observed in the second time period of this study reverts toward our baseline estimates in the longer-term once societal attention shifts away from the incessant problem of anti-Black racism. Such a scenario of a temporary shift in race relations would also be consistent with Blumer and Duster's process of collective definition, in the sense that exclusionary practices and discrimination may resume if the superordinate group once again feels their relative group position is being threatened [17]. An important avenue for future research, then, is to examine the extent to which immediate reductions in employment discrimination following protests against racial injustice are sustained over time.

Finally, whereas our design has focused on entry-level jobs and applicants with backgrounds as police officers, firefighters, and code enforcement officers, future research should move beyond these particular specifications in order to more broadly consider patterns of employment discrimination in the wake of protests against racial injustice. Per our application of group threat theory, we expect highly visible acts of police violence against Black individuals and widespread protests against racial injustice to yield at least a short-term reduction in employment discrimination against many Black applicants, not simply a reduction for Black former police officers, firefighters, and code enforcement officers [16, 17].

## Supporting information

**S1 File.**
(PDF)

## Acknowledgments

We are grateful to Neftalem Emanuel, Helen Kosc, and Annalena Wolcke for research assistance, and we thank Arun Frey for insightful comments on an earlier version of the manuscript.

## Author Contributions

**Conceptualization:** David S. Kirk, Marti Rovira.

**Data curation:** David S. Kirk, Marti Rovira.

**Formal analysis:** David S. Kirk, Marti Rovira.

**Funding acquisition:** David S. Kirk, Marti Rovira.

**Methodology:** David S. Kirk, Marti Rovira.

**Project administration:** David S. Kirk, Marti Rovira.

**Writing – original draft:** David S. Kirk.

**Writing – review & editing:** David S. Kirk, Marti Rovira.

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
