## [Decision Letter · Decision Letter 0]

29 Mar 2022

PONE-D-22-02153Do Black Lives Matter to Employers? A Combined Field and Natural Experiment of Racially Disparate Hiring Practices in the Wake of Protests against Police Violence and Racial OppressionPLOS ONE

Dear Dr. Kirk,

Thank you for submitting your manuscript to PLOS ONE. After careful consideration, we feel that it has merit but does not fully meet PLOS ONE’s publication criteria as it currently stands. Therefore, we invite you to submit a revised version of the manuscript that addresses the points raised during the review process.

Together, the three reviewers are expert in the substantive and methodological issues under investigation in your manuscript.

Reviewer 1 recommended minor revisions and detailed a few ways you might like to improve the manuscript. The first involves adding a little more reflection on how the design and analytical choices you made might have affected the results. The second is about providing some consideration of alternative interpretations of the results (e.g. the labour market may have been different in the two time periods). Like reviewer 3, reviewer 1 raises the issue of generalisation to other occupational groups.

To my mind, the results and discussion sections were concise, which is great, but also felt a little short. You could lengthen the discussion section by responding to reviewer 1’s comments there (PLOS ONE has no strict word limit).

Finally, reviewer 1 recommends adding some information on Black /White employment rates in the two periods (which would help the discussion about unemployment rates being linked to minority disadvantage), making a comparison with Quillian et al.s (I assume this is 2017, the PNAS piece) study, and providing some more descriptive statistics.

Reviewer 2 recommended straight accept and was very positive indeed.

Reviewer 3 recommended minor revisions. The recommendations cover two things. First, you might want to say something a little more explicit about the timing of the work (no reader is going to think that you started the project hoping for an exogenous shock like George Floyd, but reviewer 3 might be correct in saying you could saying something more about the purpose and timing of the study). Second, there is the issue of generalisation not only to other occupational groups (mentioned also by reviewer 1), but also in terms of timing (the murder of George Floyd was a particular moment in history, and former White police officers might have been particularly punished at such a moment). You could therefore say a little more about whether the findings might be different if one were to do the same study at a time when the exogenous shock is (for instance) not quite so strong as George Floyd and the widespread national protests that followed.

On the bases of these reviewers and my own reading of the paper, I am asking for minor revisions. This is a very interesting paper and I look forward to reading the revised version. You do not necessarily need to redraft the manuscript in each and every instance (reviewer 1 asks for more on occupational generalisibility while reviewer 3 suggests it’s so obvious it doesn’t need saying — I would recommend saying more rather than dropping it), but please do detail in a ‘response to review’ document your responses to the referee comments.

We look forward to receiving your revised manuscript.

Kind regards,

Jonathan Jackson, Ph.D

Academic Editor

PLOS ONE

Journal Requirements:

(This work was supported by the British Academy (PF19\\100020 to MR), the John Fell Fund of the Oxford University Press (to DK), and the Leverhulme Trust through the Leverhulme Centre for Demographic Science (to DK). The funders did not play any role in the study design, data collection and analysis, the decision to publish, or the preparation of the manuscript.)

Reviewers' comments:

Reviewer's Responses to Questions

**Comments to the Author**

1. Is the manuscript technically sound, and do the data support the conclusions?

Reviewer #1: Yes

Reviewer #2: Yes

Reviewer #3: Yes

2. Has the statistical analysis been performed appropriately and rigorously? 

Reviewer #1: Yes

Reviewer #2: Yes

Reviewer #3: Yes

3. Have the authors made all data underlying the findings in their manuscript fully available?

Reviewer #1: Yes

Reviewer #2: Yes

Reviewer #3: Yes

4. Is the manuscript presented in an intelligible fashion and written in standard English?

Reviewer #1: Yes

Reviewer #2: Yes

Reviewer #3: Yes

5. Review Comments to the Author

Reviewer #1: I recommend an acceptance without the need for any further revision. I reviewed an earlier version of this paper at a different journal. The authors addressed all the points I had in my earlier review. I believe this article would be a good fit for PLOS One. It is methodologically sound and makes an important contribution to the social sciences by studying what effect (if any) critical events can have in discriminatory behavior in the marketplace.

Reviewer #2: This is a well-written and highly interesting study of changes in racial discrimination before and after the murder of George Floyd in May 2020. The authors have conducted a field experiment in two waves, using the murder as an “exogenous shock” that separates the two waves. In line with their hypothesis, nicely argued along the lines of group threat theory, they find that the black/white difference in interview callbacks is reversed in wave two, suggesting an employer preference for black applicants in the immediate aftermath of the murder of George Floyd.

I definitely think that this paper merits publication. It is an original contribution to a field of research on its way to get “saturated” and the study is carefully executed, including convincing arguments for the choice of names, occupations, and the fictitious applicants’ prior professions. The analyses are also well done and presented in an accessible way. Still, I would recommend a bit more reflection on how some of the authors’ choices may affect their findings, as well as pointing out some alternative interpretations of the results that should be discussed.

The authors have made an interesting choice in letting the fictitious applicants have former work experience as police officers, firefighters, or code enforcement officers. In the supporting information document, they argue convincingly that it is not uncommon for individuals with such work experience to seek new career paths. However, the special type of prior professions begs the question of whether one would find the same effects for applicants with completely different qualifications and work experiences, or whether the change in employer behavior would be similar across different backgrounds. In other words: do the authors interpret their findings as a general reduction in hiring discrimination against black or primarily a reduction in discrimination against individuals with prior experience as police officers and fire fighters?

Relatedly, the authors point out on p. 15 that the “results may not fully generalize to the population of jobs in a labor market.” This is an understatement. All field experiments suffer from this limitation, but this one most certainly so as the four different categories of service-related jobs that are applied to represent a very small set of available jobs. I would strongly recommend that the authors acknowledge this limitation and offer some more space in reflecting on its implications.

The authors also point out that the general labor market situation characterizing the two waves was quite different because the first wave was conducted in the midst of the pandemic when the unemployment rates increased dramatically. However, I do not find that they really reflect on this problem besides arguing that a focus on black/white differences in callbacks is the most relevant outcome measure in both waves. Now, I certainly do not think that the reversal of the black/white disparities could be explained by a change in general employment levels, but some previous studies (e.g., the work of Stijn Bart) suggest that minority disadvantage tend to be higher when unemployment rates are high. Indeed, it would be beneficial if the authors added information about the average black/white employment rates in the two periods and in the areas of which the study was conducted. If the black/white difference was reduced, this would probably explain some of the effect, yet the gap was probably not turned on its head, suggesting that the study actually do document a reduction in discrimination.

The authors could also consider comparing the discrimination rates in the two separate waves to the overall discrimination rate against black applicants across time in the cited Quillian et al. study. I believe the readers would be interested in learning whether the baseline results (wave 1) align with what has already been established.

Finally, I think the paper would benefit from a table with a description of study characteristics. This information provides a useful overview of the study an makes it more transparent.

Reviewer #3: This is an excellent study with important implications and appropriate and thoroughly documented design and analysis. I recommend publication, subject to some minor revisions.

1. It would be helpful to the reader to understand how it is that the authors came to be doing a study that was interrupted by the George Floyd protest. This currently reads almost as if this was by design, but obviously this can’t be the case. What was the larger purpose of the study?

2. Consider how the use of a former police officer as a characteristic of the candidate following the case of police killing limits the generalizability of the study. For example, might it be the case that a former white police officer is particularly punished in this circumstance? The interaction models speak to this, I believe, but the authors should be more explicit about this.

3. The point about the study not generalizing to the entire labor market is true, but probably doesn’t need to be said. Does any audit study generalize to the entire labor market?

Congratulations to the authors on this excellent study.

6. PLOS authors have the option to publish the peer review history of their article (what does this mean?). If published, this will include your full peer review and any attached files.

Reviewer #1: No

Reviewer #2: No

Reviewer #3: No

---

## [Author Response · Author response to Decision Letter 0]

6 Apr 2022

We thank the reviewers and editor for their many helpful and insightful comments. We gave careful attention to all of the detailed suggestions offered by the reviewers and we believe we have made corresponding revisions in the paper to address them. Below we reproduce reviewer comments (with the exclusion of general summary comments), with our response immediately thereafter (marked by a *). Line numbers referenced below refer to those found in the unmarked version of the revised manuscript.

Response to Reviewer #1: 

Reviewer 1 recommended an acceptance. We appreciate the enthusiasm for our work. 

Response to Reviewer #2: 

I definitely think that this paper merits publication. It is an original contribution to a field of research on its way to get “saturated” and the study is carefully executed, including convincing arguments for the choice of names, occupations, and the fictitious applicants’ prior professions. The analyses are also well done and presented in an accessible way. Still, I would recommend a bit more reflection on how some of the authors’ choices may affect their findings, as well as pointing out some alternative interpretations of the results that should be discussed. The authors have made an interesting choice in letting the fictitious applicants have former work experience as police officers, firefighters, or code enforcement officers. In the supporting information document, they argue convincingly that it is not uncommon for individuals with such work experience to seek new career paths. However, the special type of prior professions begs the question of whether one would find the same effects for applicants with completely different qualifications and work experiences, or whether the change in employer behavior would be similar across different backgrounds. In other words: do the authors interpret their findings as a general reduction in hiring discrimination against black or primarily a reduction in discrimination against individuals with prior experience as police officers and fire fighters?

*Reviewer 2 has hit upon a central question. We do not wish to extrapolate beyond what is sensible with our data, so all we can reasonably say at the moment is that our findings pertain to the front-line public sector employees that are the focus of our study. However, in the Discussion section of the paper, we call for additional research using different qualifications than the ones used in this study (lines 462-470). Nevertheless, group threat theory leads us to expect that there would be a general reduction in employment discrimination in the wake of highly publicized police violence and associated protest activity.

Relatedly, the authors point out on p. 15 that the “results may not fully generalize to the population of jobs in a labor market.” This is an understatement. All field experiments suffer from this limitation, but this one most certainly so as the four different categories of service-related jobs that are applied to represent a very small set of available jobs. I would strongly recommend that the authors acknowledge this limitation and offer some more space in reflecting on its implications.

*We thank the reviewer for the suggestion to consider the implications of our sampling design with respect to generalizability and study limitations. We have added some corresponding discussion to lines 335-346.

The authors also point out that the general labor market situation characterizing the two waves was quite different because the first wave was conducted in the midst of the pandemic when the unemployment rates increased dramatically. However, I do not find that they really reflect on this problem besides arguing that a focus on black/white differences in callbacks is the most relevant outcome measure in both waves. Now, I certainly do not think that the reversal of the black/white disparities could be explained by a change in general employment levels, but some previous studies (e.g., the work of Stijn Bart) suggest that minority disadvantage tend to be higher when unemployment rates are high. Indeed, it would be beneficial if the authors added information about the average black/white employment rates in the two periods and in the areas of which the study was conducted. If the black/white difference was reduced, this would probably explain some of the effect, yet the gap was probably not turned on its head, suggesting that the study actually do document a reduction in discrimination.

*We appreciate the suggestion to reflect on the likely differences in discrimination rates in tight vs. slack labor markets. We now do so on lines 317-334. Given that discrimination tends to be greater in slack labor markets, our findings may actually understate the effect of highly publicized police violence and widespread protest activity, if it had occurred at a time not characterized by high unemployment.

The authors could also consider comparing the discrimination rates in the two separate waves to the overall discrimination rate against black applicants across time in the cited Quillian et al. study. I believe the readers would be interested in learning whether the baseline results (wave 1) align with what has already been established.

*We concur that it is useful to compare baseline rates of discrimination in our study to the trends examined in the Quillian et al. (2017) meta-analysis. Indeed, we find that the ratio of White-to-Black call-backs in our study are highly similar to those reported in the Quillian et al. study (see lines 383-391).

Finally, I think the paper would benefit from a table with a description of study characteristics. This information provides a useful overview of the study an makes it more transparent.

*We have added Table 1 showing study characteristics, namely observations by time period, applicant race, gender, and prior profession.

Response to Reviewer #3: 

1. It would be helpful to the reader to understand how it is that the authors came to be doing a study that was interrupted by the George Floyd protest. This currently reads almost as if this was by design, but obviously this can’t be the case. What was the larger purpose of the study?

*Reviewer 3 suggested we provide further detail about the original motivation of our study, which was interrupted by the murder of George Floyd. Our original study was in fact designed to examine reactions to police officers and their potential stigma in the labor market. Of course we could not have anticipated the murder of Mr. Floyd or the reaction to his death. Whereas the attention generated by the murder of Mr. Floyd was a vast order of magnitude greater than many deaths, the reviewer presumably knows that roughly 1,000 to 1,100 individuals are killed by the police each year in the United States (according to a variety of data sources, including the Washington Post police shootings database, Fatal Encounters, and mappingpoliceviolence.org). Hence, while our original intent was to study reactions to the police in the context of what might be called their more typical incidents of violence, the death of Mr. Floyd tragically provided an opportunity to examine our question in the context of widespread protests and media attention. We have sought to clarify these points on lines 170-176.

2. Consider how the use of a former police officer as a characteristic of the candidate following the case of police killing limits the generalizability of the study. For example, might it be the case that a former white police officer is particularly punished in this circumstance? The interaction models speak to this, I believe, but the authors should be more explicit about this.

*Per the reviewer’s excellent suggestion, we provide a broader discussion of the limitations of our design choices (lines 347-355), including limits on generalizing beyond the professional background characteristics examined in the study. In the Discussion section, we encourage future research that is designed in such a way to provide broader generalizability (lines 462-470).

3. The point about the study not generalizing to the entire labor market is true, but probably doesn’t need to be said. Does any audit study generalize to the entire labor market?

*While appreciate the reviewers point, but have opted to retain this particular discussion about generalizability in order to address Reviewer 2’s comments.

---

## [Editor Report · Decision Letter 1]

19 Apr 2022

Do black lives matter to employers? A combined field and natural experiment of racially disparate hiring practices in the wake of protests against police violence and racial oppression

PONE-D-22-02153R1

Dear Dr. Kirk,

We’re pleased to inform you that your manuscript has been judged scientifically suitable for publication and will be formally accepted for publication once it meets all outstanding technical requirements.

Kind regards,

Jonathan Jackson, Ph.D

Academic Editor

PLOS ONE
---

## [Editor Report · Acceptance letter]

21 Apr 2022

PONE-D-22-02153R1 

Do black lives matter to employers? A combined field and natural experiment of racially disparate hiring practices in the wake of protests against police violence and racial oppression 

Dear Dr. Kirk:

I'm pleased to inform you that your manuscript has been deemed suitable for publication in PLOS ONE. Congratulations! Your manuscript is now with our production department. 

Kind regards, 

on behalf of

Dr. Jonathan Jackson 

Academic Editor

PLOS ONE